# Investigation of Air Pocket Behavior in Pipelines Using Rigid Column Model and Contributions of Time Integration Schemes

**Arman Rokhzadi *** and **Musandji Fuamba**

Department of Civil, Geological and Mining Engineering, University of Polytechnique Montréal, C.P. 6079, Succ. Centre-Ville, Montréal, QC H3C 3A7, Canada; musandji.fuamba@polymtl.ca
* Correspondence: arman.rokhzadi@polymtl.ca

**Abstract:** This paper studies the air pressurization problem caused by a partially pressurized transient flow in a reservoir-pipe system. The purpose of this study is to analyze the performance of the rigid column model in predicting the attenuation of the air pressure distribution. In this regard, an analytic formula for the amplitude and frequency will be derived, in which the influential parameters, particularly, the driving pressure and the air and water lengths, on the damping can be seen. The direct effect of the driving pressure and inverse effect of the product of the air and water lengths on the damping will be numerically examined. In addition, these numerical observations will be examined by solving different test cases and by comparing to available experimental data to show that the rigid column model is able to predict the damping. However, due to simplified assumptions associated with the rigid column model, the energy dissipation, as well as the damping, is underestimated. In this regard, using the backward Euler implicit time integration scheme, instead of the classical fourth order explicit Runge–Kutta scheme, will be proposed so that the numerical dissipation of the backward Euler implicit scheme represents the physical dissipation. In addition, a formula will be derived to calculate the appropriate time step size, by which the dissipation of the heat transfer can be compensated.

**Keywords:** sewer network systems; surge pressure distribution; air pocket entrapment; rigid column model; implicit scheme

## 1. Introduction

The operational problems in sewer network systems, either due to overflowing or blockage at pipeline ends, disturb the boundary conditions and result in the transition of the gravity flow to a partially pressurized flow, and then the high pressurized air pocket could be released to the atmosphere either at upstream or downstream boundaries. For numerical simulation, among various models, the rigid column (RC) model is less complicated, while incorporates the fundamental features of this type of transient flows. The RC model has been studied in extensive research exemplified by the following studies [1–10]: Cabrera et al. [2], Liou and Hunt [3], Li and McCorquodale [4], Zhou et al. [5,6], Lee [7], Fuertes-Miquel et al. [8], Coronado-Hernández et al. [9,10].

Li and McCorquodale [4] used the RC model proposed in [1] to calculate the transient pressure which causes blowing off of storm manholes. In the proposed RC model, the water column is assumed as an incompressible fluid with uniform but unsteady velocity. The transition from free-surface flow to pressurized flow was described in six stages, and the trapped air pocket was assumed to undergo pseudo adiabatic expansion and compression. It was found that the numerical transient pressure overpredicts the experimental data by a factor of two, and the attenuation of the pressure oscillation is underestimated, while the frequency of pressure surge oscillations predicts the measured data accurately. Li and McCorquodale [4] linked this poor behavior to the superposition of various air release processes and the steady-state friction factor used in the mathematical model.

Liou and Hunt [3] proposed an RC model to simulate the transient flow in an empty pipeline with an undulating elevation profile. They found that the calculated velocity distribution is comparable with the experimental data.

Lee [7] analytically and experimentally studied the behavior of air pressurization in horizontal pipelines and derived a closed form of the solutions calculated by the RC model, in which a vertical air-water interface is assumed. It was found that the maximum pressure increases when the reservoir pressure increases, and the RC model overpredicts the experimental data. In addition, Lee showed that the damping behavior of the air pressure is contributed to the heat transfer between the air, water, and the pipe wall. Additionally, he showed that the maximum pressure calculated by both variable and constant water length assumptions is similar, while he indicated that these two assumptions lead to calculating different times at which the maximum pressure occurs. Lee demonstrated that the maximum pressure in a frictionless pipe is independent of initial air and water lengths. Moreover, Lee, by using the small wave theory, linearized the governing equations and provided a formula for the frequency of the air pressure oscillation. This formula showed that the frequency depends on the reservoir pressure and the initial length of the air and water phases.

Hou et al. [11] experimentally and numerically studied the rapid filling of a large-scale pipeline. For the numerical calculation, they applied the RC model proposed in [3] along with the modification proposed by Axworthy and Karney [12]. Note that the modification is to eliminate the pressure head at junctions from the momentum equation. They showed that the numerical result of the RC model is in good agreement with the experimental data.

Fuertes-Miquel et al. [8], using the RC model, analyzed the effects of isothermal and adiabatic behaviors of the air pocket, which is expelled from a pipe through an air valve by admitting water flow. It was found that numerical results can be significantly different depending on the heat transfer mechanism. In particular, in contrary to the isothermal assumption, the adiabatic assumption results in smaller air pressure and larger water velocity so that the induced water hammer is greater.

Coronado-Hernández et al. [9] studied the sub-atmospheric pressure occurring during an emptying process in a pipeline with irregular elevation. They proposed an RC model to simulate the water flow, while the air pocket was simulated using the thermodynamic formulation. By comparing to experimental data, they showed that the RC model can predict the flow variables (water velocity and air pressure) accurately.

The air entrapment following rapidly filling storm water systems (SWSs) has great complexity and challenges. This complexity, for example, includes the transition from gravity to pressurized flow and their interaction and significant head loss at the interface, the turbulent nature of the flow, lumped nature of the air pocket [5], and the heat transfer between water, air, and the pipe wall [7]. It is known that the available mathematical models, which practitioners are interested in, including the RC model, do not take the effects of all of these parameters into account so that these models overestimate the numerical solutions including the maximum pressure. This overestimation implies that these mathematical models, due to their assumptions, are not able to sufficiently predict the dissipation of the driving energy. Thus, to dissipate this energy, or in other words, to compensate for the effects of the aforementioned neglected terms, considering a numerical dissipation term seems to be useful. Note that in the literature, the classical fourth order Runge–Kutta explicit scheme has mostly been employed for the time integration of the governing equations. Recently, the backward Euler implicit scheme has been used in [13] to simulate the dynamics of the entrapped air pocket using the 3D Navier–Stokes equation and the volume of fluid (VOF) method. In addition, Rokhzadi and Fuamba [14] proposed using the backward Euler implicit scheme to solve a similar problem by a shock-fitting approach.

In this paper, the damping and frequency of the pressure distribution of an entrapped air caused by a rapid filling of a horizontal reservoir-pipe system will be studied using some available experimental data as well as the numerical results of the RC model. In this regard, the effects of reservoir pressure and the air and water initial lengths will be

discussed. To analytically show the relation between these parameters and the damping and frequency, the governing equations are linearized around a point, in which the air pressure is equal to the reservoir pressure. This linearization implies that the air pressure does not significantly deviate from the reservoir pressure. Although this assumption is not valid for every case, it can provide insightful information about the damping and frequency. In addition, using the backward Euler implicit scheme will be introduced to discuss the numerical dissipation as a representative for the physical dissipation, which the RC model cannot predict properly for many reasons. In this regard, the modified equation will be derived to provide a formula, by which the size of the time step and the amount of numerical dissipation can be controlled so that excessive numerical dissipation imposed on the solutions is avoided. It will be shown that this formula adjusts the time step in proportion to the values of the effective physical parameters on the damping behavior.

It is worth mentioning that for problems with large initial air pocket sizes, the elasticity of entrapped air pocket is much higher than the elasticity of the water [15]. Thus, it can be expected that for these cases, which are more problematic in SWSs, the RC model can calculate the solutions appropriately. However, as indicated in the literature, e.g., [16], the RC model cannot capture the interactions between different waves exist in pipelines.

## 2. Numerical Analysis

### 2.1. Governing Equation

In this paper, the air entrapment caused by transient flow in a horizontal reservoir-pipe system is studied. As shown in Figure 1, a pipe segment, with the diameter D and length $L_t$, is connected at upstream to a reservoir, in which the absolute pressure is $p_R$, and is blocked at downstream by a valve. The water in a stationary pressurized flow regime occupies an upstream part of the pipe with the initial length L. The entrapped air occupies the rest of the pipe at downstream, which is separated by a valve from the water column, with the initial length $L_a$, which $L_a = L_t - L$. Note that the interface of air–water phases is assumed vertical so that the gravity flow forming during the transient flow is neglected.

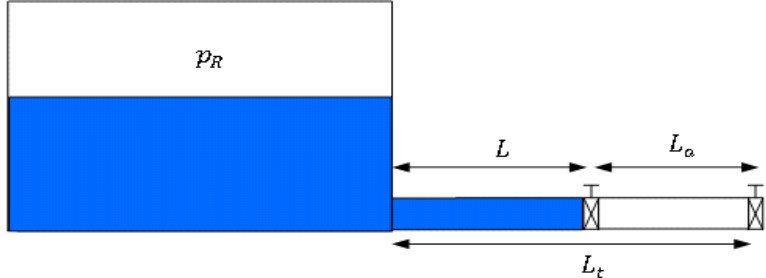

**Figure 1.** The schematic of a horizontal reservoir-pipe system with constant driving pressure.

The momentum equation for a horizontal water column is,

$$\frac{dV}{dt} = \frac{(p_R - p)}{\rho L} - \frac{f}{2D}V|V|, \tag{1}$$

where t is the time variable, $\rho$ and V are the water density and velocity, respectively, and g is the gravitational constant (g = 9.81 ms$^{-2}$), p is the air absolute pressure, and f is the friction factor. Note that the length of the water column is assumed constant as Lee [7] demonstrated that the variable length does not provide significantly different results compared to the constant length assumption.

The continuity equation gives a relationship between the air volume rate and the water flow rate as,

$$\frac{d\forall}{dt} = -VA, \tag{2}$$

where A is the pipe cross-section area and $\forall$ is the air volume. Note that the velocity is assumed positive in the downstream direction. Following Lee [7], applying the first thermodynamic law to the air pocket and assuming the air as a perfect gas yield,

$$\frac{dp\forall}{dt} = -(k-1)p\frac{d\forall}{dt},\tag{3}$$

where k is the polytropic exponent. Note that Equation (3) implies a balance between the internal energy and the work done through the air compression and expansion. However, Lee [7] attributed the damping of the air pressure distribution to the heat transfer. Thus, to consider the effect of heat transfer, the term $(k-1)q$ must be added to the right hand side (RHS) of Equation (3). In addition, in the present study, the polytropic exponent is set to $k = 1.4$, as indicated in [7], meaning that this value gives more appropriate results. Substituting Equation (2) into Equation (3) yields,

$$\frac{dp\forall}{dt} = (k-1)pVA.\tag{4}$$

*2.2. Linearized Governing Equations*

As indicated in [7], the variable water column length does not have significant effects on the solutions. Thus, the only nonlinearities in the governing equations are related to the friction force in the momentum equation and the polytropic process equation $(p^k = cte)$. As shown in [7], for small ranges of reservoir pressure $(p_R < 2.0p_a)$, in which $p_a$ is the atmospheric absolute pressure $(p_a = 101\ kPa)$, the maximum air pressure, so as the minimum air pressure, slightly changes with the friction loss coefficient $(fL_a/D)$ and it is almost constant for both polytropic coefficients $(k = 1.2\ and\ 1.4)$. Thus, for simplicity of the discussion presented only in Sections 2.2 and 2.3, the friction and local loss terms are neglected, meanwhile, the contributions of these terms to the damping behavior of the entrapped air pressure have been discussed in the literature.

Thus, the linearized governing equations are expected to appropriately approximate the solutions of the nonlinear equations when $p_R/p_a < 2.0$. Therefore, the governing equations are linearized around a point, in which $p = p_R$, so that the effects of different variables on the damping and frequency of the air pressure distribution can be analyzed.

A continuous function of two variables $f(u, v)$, in which u and v denote the dependent variables, could be linearized using the Taylor series expansion around any point $(u_R, v_R)$ as,

$$f(u, v) = f(u_R, v_R) + \left.\frac{\partial f}{\partial u}\right|_{(u_R)}(u - u_R) + \left.\frac{\partial f}{\partial v}\right|_{(v_R)}(v - v_R).\tag{5}$$

Note that the momentum equation, Equation (1), and the continuity equation, Equation (2), for a frictionless horizontal pipe are already linear. The energy conservation equation, Equation (4), can be linearized using Equation (5) as,

$$\frac{dp}{dt} = kA\frac{p_R}{\forall_R}V + (k-1)\frac{AV_R}{\forall_R}(p - p_R),\tag{6}$$

where the subscript (R) implies the value of a variable when the air pressure equals to the reservoir pressure.

*2.3. Pressure Distribution Analysis*

The linearized governing equations, Equations (1) and (6), can be rearranged in a vector form as,

$$\frac{d\vec{F}}{dt} = \mathbf{G}\,\vec{F},\tag{7}$$

where,

$$\vec{F} = \left\{ \begin{matrix} V \\ p^* \end{matrix} \right\}, \mathbf{G} = \left( \begin{matrix} 0 & -1/\rho L \\ kAp_R/\forall_R & (k-1)AV_R/\forall_R \end{matrix} \right), \tag{8}$$

where $p^* = p - p_R$. Two time-dependent decoupled equations can be derived using the Eigen-decomposition method, by which the numerical solutions, which have an exponential form in terms of the time, can be found. The amplitude and the frequency of the solutions are related to the eigenvalues of the matrix G, which can be calculated as,

$$\lambda_{1,2} = \frac{1}{2} \left\{ -(k-1)\left(\frac{p_R}{p_a}\right)^{1/k}\frac{V_R}{L_a} \pm i\sqrt{4k\frac{1}{L_aL}\frac{p_R}{\rho}\left(\frac{p_R}{p_a}\right)^{1/k} - \left((k-1)\left(\frac{p_R}{p_a}\right)^{1/k}\frac{V_R}{L_a}\right)^2} \right\} \tag{9}$$

where i denotes the complex unit number. Note that the polytropic process relationship, i.e., $p^k = cte$, and the definition of $= AL_a$ were used to further rearrange Equation (9). Thus, the amplitude and the frequency of the air pressure distribution can be calculated as,

$$M = e^{-\frac{1}{2}\left((k-1)\left(\frac{p_R}{p_a}\right)^{1/k}\frac{V_R}{L_a}\right)}, \tag{10}$$

$$\varphi = \sqrt{4k\frac{1}{L_aL}\frac{p_R}{\rho}\left(\frac{p_R}{p_a}\right)^{1/k} - \left((k-1)\left(\frac{p_R}{p_a}\right)^{1/k}\frac{V_R}{L_a}\right)^2}, \tag{11}$$

where $\varphi$ is the frequency and M is the amplitude. Note that the damping behavior can be realized from the amplitude (M) in Equation (10). As explained, $V_R$ denotes the water column velocity at a time when the air pressure is equal to reservoir pressure. Malekpour et al. [17] analyzed the energy conversion of a transient flow in a reservoir-pipe system and derived an equation for the water column velocity in terms of initial air and water column lengths, reservoir pressure, and the air pressure. However, a simple equation for the cases with constant water column length assumption is derived here.

Multiplying both sides of Equation (1), in which the friction term is neglected, with VA and summing up with Equation (4) yields an energy equation as,

$$\frac{d}{dt}\left(\rho LA\frac{V^2}{2} + \frac{p\forall}{k-1}\right) = p_RAV. \tag{12}$$

For an initially stationary water column, taking integral of both side of Equation (12) from initial condition to the time when the pressure is equal to the air pressure, so called $t_R$, yields,

$$\rho LA\frac{V_R^2}{2} + \frac{p_R\forall_R}{k-1} - \frac{p_0\forall_0}{k-1} = \int_0^{t_R} p_RAVdt. \tag{13}$$

Since the time taken by the air pressure to reach the reservoir pressure is not so long, for initially stationary flow, the integral on RHS of Equation (13) is negligible. Thus, the following relation for $V_R$ can be derived,

$$V_R^2 = \frac{2p_RL_a\left(p_0 - (p_0/p_R)^{1/k}\right)}{\rho L(k-1)}. \tag{14}$$

As can be seen, $V_R$ is a function of $\sqrt{p_RL_a/L}$. Note that a similar function can be found in [17] (Equation (20)). Therefore, Equations (10) and (11) show that the frequency and the damping of the air pressure distribution increase as the reservoir pressure increases and as the product of the air and water lengths ($L_aL$) decreases. Later, these relationships will be confirmed by explaining the results of some available experiments. Therefore, Equation (10) shows that the RC model is able to predict the damping behavior of the air pressure distribution. However, in the next section, it will be shown that this ability depends on the time integration scheme.

### 2.4. Time Integration Scheme

From the literature, it can be realized that commonly the classical fourth order Runge–Kutta scheme, hereafter called the 4th-RK scheme, has been mostly used in the literature to integrate the governing equations of the RC model [6,11]. In addition, it is known that the numerical results are always overestimated, caused by underestimating the physical energy damping for many reasons including the assumption of lumped air pocket process [5], and neglecting the heat transfer between the air, water, and pipe wall [7]. Therefore, this paper proposes using the backward Euler implicit time integration scheme, hereafter called the BE scheme, which has more dissipative properties compared to 4th-RK scheme. In other words, numerical dissipation can represent physical dissipation.

The governing equations of the RC model for a frictionless horizontal pipe-reservoir system are Equation (1), without the last term in its RHS, and Equations (2) and (3). However, Equation (3) can be further simplified, using the continuity equation in Equation (2), as,

$$\frac{dp}{dt} = k\frac{p}{\forall}AV. \tag{15}$$

By applying the BE scheme to the governing equations, the discretized form of the equations become,

$$V^{n+1} = V^n + \Delta t\frac{\left(p_R - p^{n+1}\right)}{\rho L}, \tag{16}$$

$$\forall^{n+1} = \forall^n - \Delta t A V^{n+1}, \tag{17}$$

$$p^{n+1} = p^n + \Delta t k\frac{p^n}{\forall^n}AV^{n+1}, \tag{18}$$

where superscript n implies the current time step. Note that in Equation (18) the term $p/$ can also be calculated implicitly. However, it was realized that it does not have significant effects on the results. The details of the classical fourth order Runge–Kutta scheme can be found in other references, e.g., [18].

### 2.5. Effective Time Step Size

Rokhzadi and Fuamba [14], for problems with small ratios of the reservoir pressure to the atmospheric pressure, used a shock-fitting approach, which is a combination of the RC model and the Saint-Venant equations. Rokhzadi and Fuamba proposed using the BE scheme instead of the 4th-RK scheme. In this shock-fitting approach, the time step size is controlled by the Courant Friedrichs Lewy (CFL) condition associated with the gravity flow. Since by using the RC model there is no such criterion to control the time step size, so as the numerical dissipation, the question is how much numerical dissipation can be allowed on the calculation to avoid excessively imposing numerical dissipation and to prevent spoiling the solutions. In this section, a formula will be derived, by which it will be shown that the time step size is controlled by the physical variables, which affect the damping.

In order to find the numerical dissipation terms imposed by the BE scheme, the modified equation is needed. The modified equation can be found using Equations (16)–(18) and by substituting the Taylor series of the variables at the time step n + 1. Note that here, only those terms with $O(\Delta t)$ are kept and the terms with the higher order of $\Delta t$ are neglected.

$$\frac{dV^n}{dt} = \frac{p_R - p^n}{\rho L} - \Delta t\frac{1}{\rho L}\frac{dp^n}{dt} + O\left(\Delta t^2\right), \tag{19}$$

$$\frac{dp^n}{dt} = k\frac{p^n}{\forall^n}AV^n + \Delta t k A\frac{p^n}{\forall^n}\frac{dV^n}{dt} + O\left(\Delta t^2\right). \tag{20}$$

By comparing Equations (19), and (20) to Equation (1), without the friction term, and Equation (15), it can be realized that the first terms in RHS of Equations (19) and (20) are the physical terms involved in the governing equations and the second terms are

the truncation error terms, which are responsible for imposing numerical dissipation. Substituting Equation (19) into Equation (20) yields,

$$\frac{dp^n}{dt} = k\frac{p^n}{\forall^n}AV^n + \Delta t k A \frac{p^n}{\forall^n}\left(\frac{P_R - p^n}{\rho L} - \Delta t \frac{1}{\rho L}\frac{dp^n}{dt} + \dots\right). \tag{21}$$

As mentioned before, the energy equation, in which the effect of heat transfer is included, has a form as,

$$\frac{dp}{dt} = k\frac{p}{\forall}AV + (k-1)\frac{q}{\forall}. \tag{22}$$

By comparing Equation (21) to Equation (22), an equation can be derived for the time step as,

$$\Delta t = \frac{\rho L(k-1)q}{kpA(p_R - p)} \tag{23}$$

Note that since Equation (23) was found by equating the heat transfer term to the numerical dissipation term, thus, these analyses imply that by applying the BE scheme to the governing equations of the RC model and using the time step size as in Equation (23), the physical dissipation of the heat transfer can be compensated by the numerical dissipation of the BE scheme. In addition, later, this compensation will be further explained. To clarify Equation (23), the heat transfer term needs to be further explained.

Following Lee [7], by neglecting the conduction and radiation heat transfers, the convection heat transfer can be calculated as,

$$q = HA_q\Delta T = HA_q(T_0 - T), \tag{24}$$

where H is the convection heat transfer coefficient, $A_q$ is the thermal area, $T_0$ is the initial air pocket temperature, and T is the variable air temperature, both in the Kelvin scale. Following [7,19], the convection coefficient is calculated as,

$$H = 3.5|\Delta T|^{1/3}. \tag{25}$$

The thermal area for the typical examples depicted in Figure 1 can be calculated as,

$$A_q = 2A + \pi DL_a, \tag{26}$$

so that the heat transfer between the air, water, and pipe wall can be taken into account.

The air temperature is calculated using the perfect gas state relationship as,

$$T = \frac{P\forall}{Rm_g}, \tag{27}$$

where R is the air constant (R = 287 J/kg·K) and $m_g$ is the air mass, which is constant since there is no air release.

Equation (23) can be further expanded, using Equations (24)–(28), as,

$$\Delta t = \frac{\rho L(k-1)HA_q\Delta T}{kpA(p_R - p)}. \tag{28}$$

As can be seen in Equation (28), the time step size depends on physical variables including the air and water initial lengths, the reservoir pressure, and heat transfer.

## 3. Results

To further discuss the effects of time integration schemes on the air pressure distribution and to clarify the effects of the reservoir pressure ($p_R$) and the product of the air and water initial lengths ($L_aL$) as well as to compare the performances of the BE and 4th-RK schemes, some test cases, for which the experimental data exist, and a hypothetical example

have been solved by the RC model, for which the governing equations are Equations (1), (2) and (15).

### 3.1. Hypothetical Test Case

A horizontal frictionless pipe-reservoir system with $D = 1$ m and different air and water lengths and the reservoir pressure ratio $p_R/p_a = 2.0$ has been solved and the results are shown in Figures 2 and 3. Note that for both BE and 4th-RK schemes, the time step is set to $\Delta t = 0.001$ s.

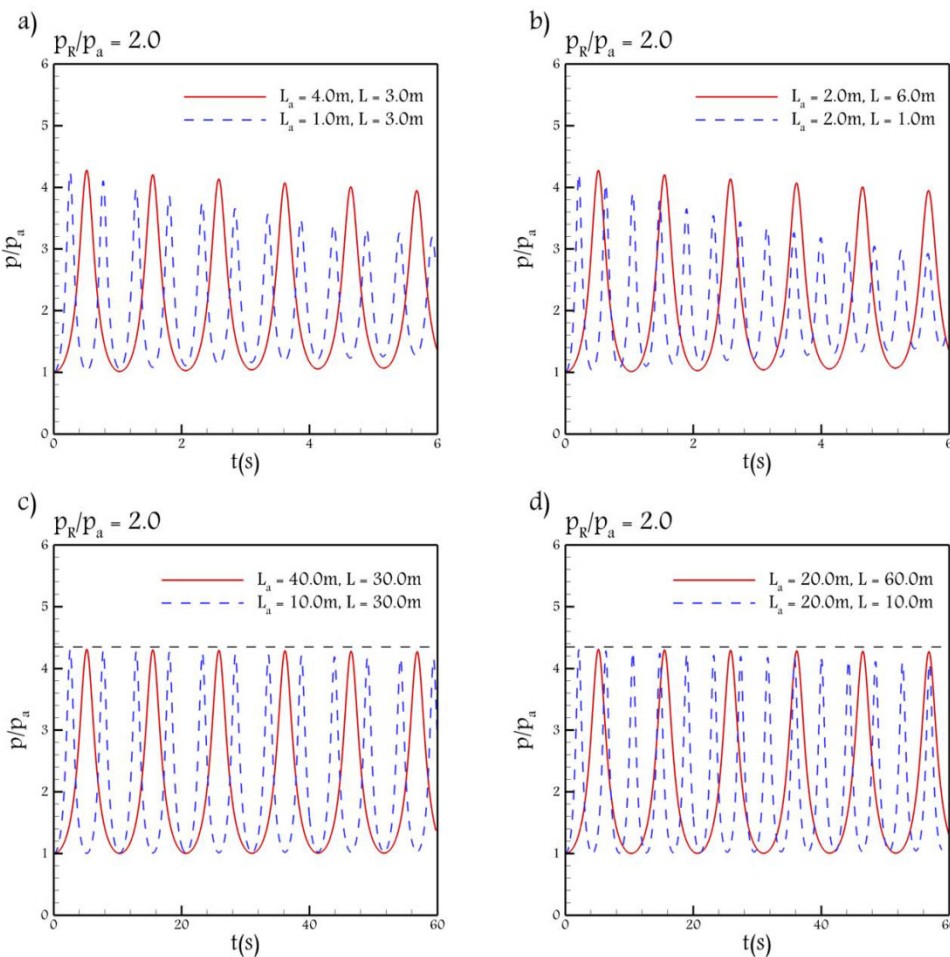

**Figure 2.** The non-dimensional air absolute pressure $(p/p_a)$ distribution vs. the time, calculated by the backward Euler scheme, for the reservoir pressure ratio $(p_R/p_a)$ of 2.0, and (**a,c**) different air initial lengths, (**b,d**) different water initial lengths.

As can be seen in Figure 2, as the product of the air and water lengths decreases, either by decreasing the air length or the water length, the rate of damping and the frequency increase. In addition, as indicated in [7] as well, Figure 2 shows that the first maximum pressure, for a frictionless pipe, is independent of the air and water initial lengths. Note that in Figure 2, it was attempted to examine large lengths of the air and columns as well (second row). Figure 3 shows the results of the same test case, in which the 4th-RK scheme is used instead of the BE scheme. As can be seen, by using the 4th-RK scheme, the RC model is unable to simulate the relationship between the damping and the product of the air and water initial lengths. However, the frequency and its relation with $L_aL$ are properly predicted.

Similarly, to illustrate the effect of the reservoir pressure on the damping and the frequency and to compare the effect of time integration schemes, Figure 4 shows the non-dimensional air pressure distribution $(p/p_a)$ in terms of time for a specific example with

pipe diameter D = 1.0 m, and the air and water initial lengths $L_a$ = 2.0, and L = 1.0 m. The left graph illustrates the results calculated by the BE scheme, and the right graph is calculated by the 4th-RK scheme. As can be seen, the frequency increases with increasing the reservoir pressure, the feature that appropriately is predicted by both time integration schemes. However, from the right graph, it can be seen that the 4th-RK scheme is unable to appropriately calculate the damping and its direct relation with the reservoir pressure, while the BE scheme in the left graph effectively calculates the damping behavior. In addition, as shown in [7], the first maximum pressure, for initially stationary water flows, is only a function of the reservoir pressure and it increases with increasing the reservoir pressure, a fact that is calculated by both time integration schemes.

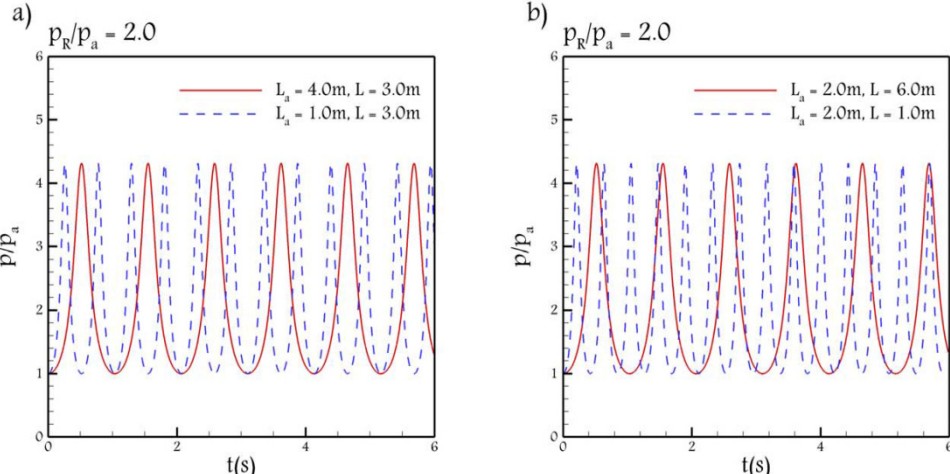

**Figure 3.** The non-dimensional air absolute pressure $(p/p_a)$ distribution vs. the time, calculated by classical 4th order Runge–Kutta scheme, for the reservoir pressure ratio $(p_R/p_a)$ of 2.0, and (**a**) different air initial lengths, (**b**) different water initial lengths.

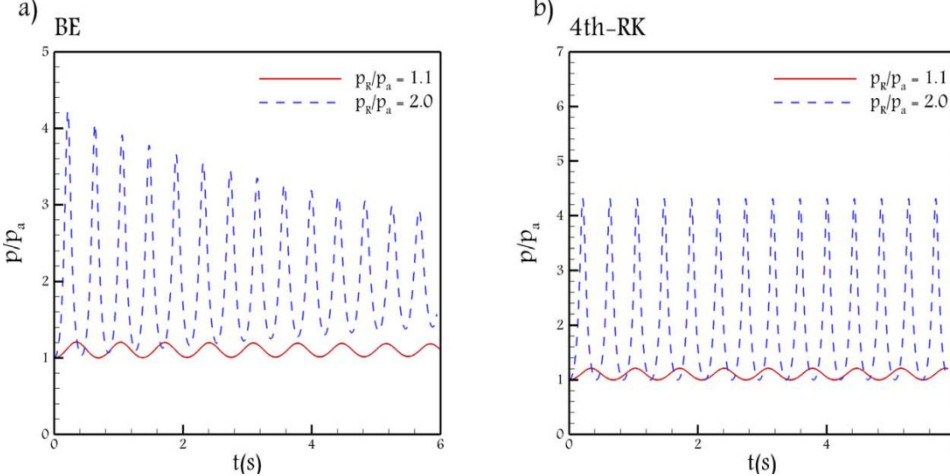

**Figure 4.** The non-dimensional air absolute pressure $(p/p_a)$ distribution vs. the time, calculated by (**a**) the backward Euler scheme, and (**b**) the classical 4th order Runge–Kutta scheme, for different reservoir pressure ratio $(p_R/p_a)$ and $L_a$ = 2.0 and L = 1.0 m.

Therefore, it can be claimed that implicit time integration schemes, represented by the BE scheme, is more appropriate time integration scheme for using with the RC model.

### 3.2. Experiment of Zhou

Note that, hereafter, in every test case, the time step is calculated using Equation (28). One of the test cases is of Zhou [20], which is a reservoir-pipe system, in which D = 0.035 m,

$L_t = 8.96$ m, $p_R/p_a = 2.43$, and $\lambda = 0.56$, and 0.89, in which $\lambda = L/L_t$. In addition, the pipe is horizontal with a friction factor $f = 0.035$ and a local head loss coefficient $K_{loss} = 0.093$. Figure 5 shows the air pressure distribution calculated by the 4th-RK and BE schemes and the corresponding experimental data for both $\lambda$'s.

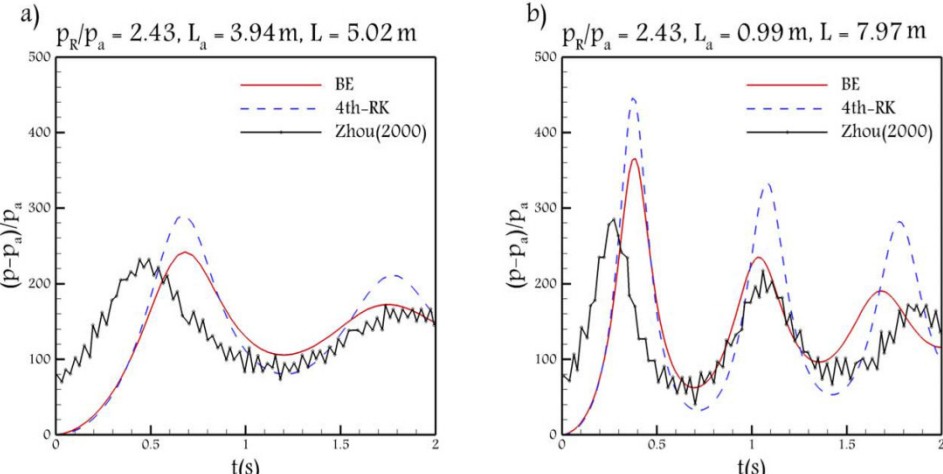

**Figure 5.** The non-dimensional air absolute pressure $(p/p_a)$ distribution vs. the time, calculated by both backward Euler scheme, and classical 4th order Runge–Kutta scheme, for a test case of Zhou [20], $p_R/p_a = 2.43$ and (**a**) $\lambda = 0.56$, and (**b**) $\lambda = 0.89$.

First, it is shown that the physical solution approves the inverse relation between the damping and the product of the air and water initial lengths. The experimental data in Figure 5a shows that the first peak with the value of 230 occurs at $t = 0.45$ s, and the second peak value of 164 occurs at $t = 1.83$ s. Thus, the rate of damping in this test case is around 47.83 $s^{-1}$. The same calculation in Figure 5b shows that the rate of damping is 102.5 $s^{-1}$. These values show that when $L_a L$ is smaller, as in Figure 5b, the rate of damping increases. Therefore, it can be claimed that the RC model and the relation derived for the amplitude of the oscillations in Equation (10) provide appropriate information about the physical behavior of the air pressurization.

In addition, the numerical solutions in Figure 5 show that the BE scheme outperforms the 4th-RK scheme. It is worth mentioning that the phase shift between numerical and experimental data was reported in other references, e.g., [5,6]. As Zhou et al. [5] explained, this phase shift could be due to the assumption that the air pocket remains intact, while the air roll ups and splits into several pockets with smaller sizes. However, for the test case with a smaller initial air length (Figure 5b), in the last period, the BE scheme causes a more obvious phase shift, compared to the 4th-RK scheme.

Here, it is shown that the numerical dissipation is imposed in proportion to physical conditions so that the performance of the RC model is improved. As demonstrated in [7], the dimensional analysis shows that the governing equations of the RC model have a similarity form with a scale factor $fL_a/D$, which appears as a coefficient of the friction loss term. The details of the non-dimensional variables can be found in Equations (3)–(85) in reference [7]. Thus, since the only differences between the test cases of Zhou [13], as in Figure 5, are the initial air and water lengths, if the coefficient $fL_a/D$ set equal for both cases, then the relation between the numerical dissipation imposed on the solutions and the product of the air and water initial lengths can be analyzed quantitatively. Therefore, the friction factor in the test case with $\lambda = 0.56$ is changed to $f = 0.035/4.0$, and the result of this test case along with Figure 5b, are shown in Figure 6.

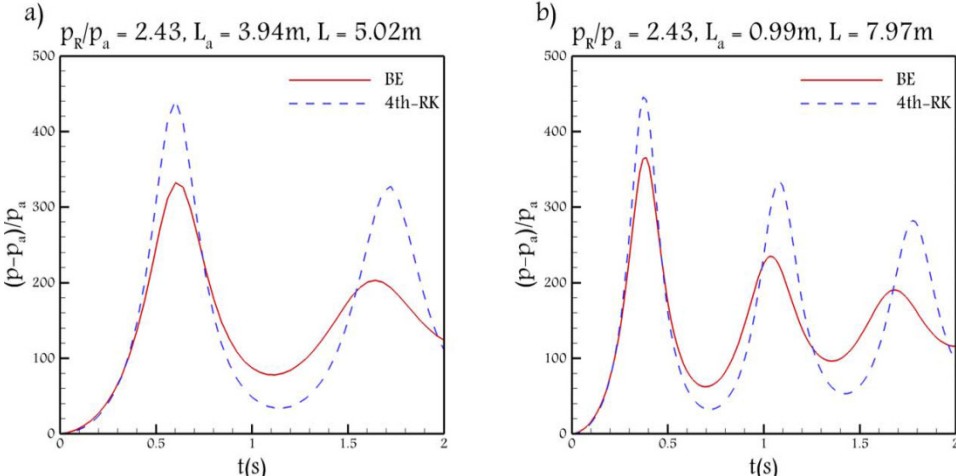

**Figure 6.** The non-dimensional air absolute pressure $(p/p_a)$ distribution vs. the time, calculated by both backward Euler scheme, and classical 4th order Runge–Kutta scheme, for a test case of Zhou [20], $p_R/p_a = 2.43$ and (**a**) $\lambda = 0.56$, and (**b**) $\lambda = 0.89$, both with the same value of $fL_a/D$.

The first peak calculated by the 4th-RK and BE schemes, for the test with $\lambda = 0.56$, shown in Figure 6a, are approximately 440, and 331, with a difference of around 109. The same calculation for the case with $\lambda = 0.89$, in Figure 6b, shows that the first peak calculated by the 4th-RK and BE schemes are approximately 445 and 362, with a difference around 83. These values show that the numerical dissipation imposed on the solutions of the test case with $\lambda = 0.56$, in which $L_aL$ is larger, is larger than the test case with $\lambda = 0.89$, in which $L_aL$ is smaller. Referring to Equation (15), it can be claimed that the BE scheme can improve the performance of the RC model because, according to Equation (15), when $L_aL$ is large the damping of the RC model is small and the overestimation is large and, as seen in Figure 6a, the BE scheme imposes more dissipation to improve the overestimation.

### 3.3. Experiment of Zhou and Liu

Another experiment presented in this paper was carried out in [21], in which the air pressurization in a horizontal reservoir-pipe system was experimentally analyzed for different tailwater depths including the initially dry-bed condition. The experiment setup is similar to the schematic provided in Figure 1, with $D = 0.04$ m, $L_t = 8.824$ m, $L_a = 3.25$ m, and $p_R = p_a + 120$ kPa. The friction factor was calculated as $f = 0.075$, and, in the present paper, an additional factor representing the local head loss is considered as $K_{loss} = 0.093$. The air pressure distribution of this test case, calculated by both BE and 4th-RK schemes, are shown in Figure 7a. Note that for this test case, $k = 1.2$ was used since it provides more accurate results than $k = 1.4$.

The experimental data in Figure 7a shows that the first peak with the value of 172 occurs at $t = 0.8$ s, and the second peak value of 142 occurs at $t = 2.0$ s. Thus, the rate of damping in this test case is around $25.0$ $s^{-1}$. Note that the product of air and water initial lengths in the test case is almost the same as the test case of Zhou [20], in which $\lambda = 0.56$ and the rate of damping was calculated as $47.83$ $s^{-1}$. Comparing the rate of damping in these two test cases shows that when the reservoir pressure is large, as in the test case of Zhou [20], the rate of damping is large as well. Therefore, the physical solution proves a direct relation between the damping and the reservoir pressure. In addition, it can be claimed that the RC model and the relation derived for the amplitude of the oscillations in Equation (10) provide appropriate information about the physical behavior of the air pressurization.

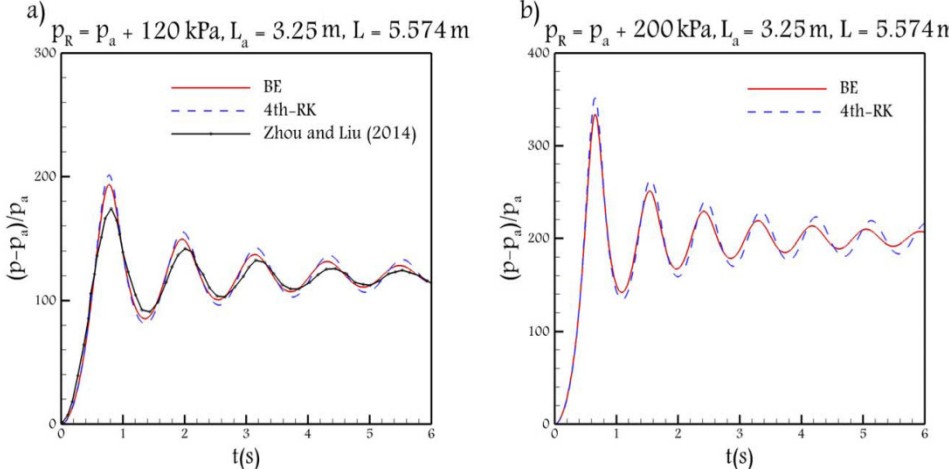

**Figure 7.** The non-dimensional air gauge pressure $(p - p_a)/p_a$ distribution vs. the time, calculated by both backward Euler scheme, and classical 4th order Runge–Kutta scheme, (**a**) for a test case of [21], $p_R = p_a + 120$ kPa and (**b**) $p_R = p_a + 200$ kPa.

Moreover, it can be seen that the BE scheme improves the overestimated peak values compared to the 4th-RK scheme and no obvious phase shift is seen.

In order to further analyze the relation between numerical dissipation and the reservoir pressure, in Figure 7b, the reservoir pressure is increased to $p_R = p_a + 200$ kPa. The difference between the first peak value calculated by BE and 4th-RK schemes in Figure 7a, in which the reservoir relative pressure is 120 kPa, is around seven. In Figure 7b, in which the reservoir relative pressure is around 200 kPa, the difference between numerical peak values is around 18. As the only difference between Figure 7a,b, is the reservoir pressure, it can be claimed that for cases with larger reservoir pressure, in which the overestimation of the RC model is larger, the BE scheme imposes more dissipation to improve the overestimation.

Therefore, considering the results of both test cases, it can be expected that implicit time integration schemes, represented by the BE scheme, are more helpful in using with the RC model. In addition, by using the derived formula for the time step, it can be ensured that excessive numerical dissipation is avoided.

### 3.4. Test Case of Lee

In this section, to show that the BE scheme by using the time step derived in Equation (28) can compensate for the effect of heat transfer, one of the test cases solved in [7] by 4th-RK scheme, in which the effect of heat transfer is included is compared to the result calculated by BE without the heat transfer. The test case is a frictionless horizontal reservoir-pipe system with $D = 1.0$ m, the initial water length $L = 40.0$ m and the initial air length $L_a = 10.0$ m. The ratio of the reservoir absolute pressure to the atmospheric pressure is $p_r/p_a = 2.0$. Note that following Lee [7], the variable water column length is considered in the RC model. As can be seen in Figure 8, the result calculated by BE scheme is almost the same as the result calculated in [7]. Thus, it can be claimed that the numerical dissipation of the BE scheme, by using the time step in Equation (28), can compensate for the effect of heat transfer.

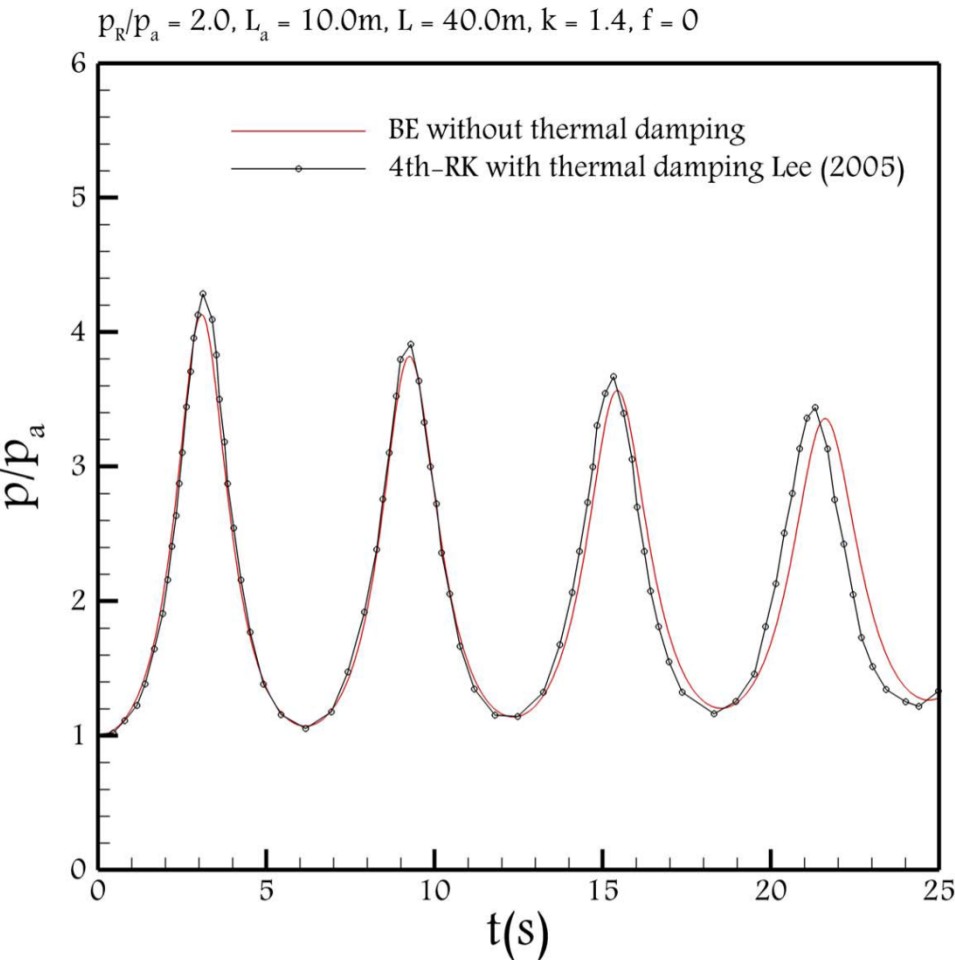

**Figure 8.** The non-dimensional air absolute pressure $(p/p_a)$ distribution vs. the time, calculated by backward Euler scheme without thermal damping, and the result calculated in [7] and illustrated in [7] (Figure 3.10) by 4th order Runge–Kutta scheme with thermal damping, $\frac{P_R}{P_a} = 2.0$.

## 4. Conclusions

This study analyzes the rigid column model to further explain that this model is able to produce the fundamental features of the physics of the problem, in particular, the damping behavior of the surge pressure distribution. However, this model underestimates the damping behavior and overestimates the peak values. The reasons are linked to simplified approximations associated with the rigid column model and the complexity of the transient flow, which cause this model to underestimate the prediction of the physical energy dissipation. To address this poor behavior of the rigid column model, using implicit schemes has been proposed, by which the numerical dissipation can represent the physical dissipation. In this regard, a criterion has been provided to control the amount of numerical dissipation to ensure that the solutions are not spoiled and, in the meantime, the performance of the rigid column model is improved.

The contribution of the friction loss term to the damping behavior of the surge pressure is known and discussed in the literature. Therefore, to provide an analytic formula by which the relation between the frequency and damping and physical parameters can be shown, the governing equations, applied to a frictionless horizontal reservoir-pipe system, were linearized around the point where the air pressure is equal to the reservoir pressure. This linearization implies that the air pressure does not deviate significantly from the reservoir pressure. Although this assumption is not valid for every example, it was shown that it can provide insightful information about the air pocket behavior. It should be noted that the linearized governing equations were just used to analyze the influential factors

on the damping and frequency, while for solving the test cases, the nonlinear governing equations were used.

It was found that the damping and frequency of the air pressure, calculated by the RC model, has a direct relation with the reservoir pressure and inverse relation with the product of the air and water initial lengths. The same relationship was found in some available experimental data, meaning that it confirms that the RC model is able to predict the physical behavior of the air pressure distribution. In addition, it was shown that, despite the abilities of the RC model, the performance of this model depends on the type of the temporal scheme used for time integration, which is because of many simplified approximations associated with the RC model, including neglecting the heat transfer and lumped assumption of the air pocket. In this regard, it was shown that implicit schemes, represented by the backward Euler scheme, are more effective than explicit schemes, represented by the classical fourth order Runge–Kutta scheme. The reason is that the numerical dissipation associated with implicit schemes can compensate for the physical dissipation associated with the neglected factors in the RC model. Moreover, to avoid imposing excessive numerical dissipation on the solutions, a formula for calculating the time step size was derived, which is a function of the effective parameters on the physical damping and frequency. It was shown that, by using this time step, the backward Euler scheme can help the RC model to more effectively predict the air pressure distribution.

**Author Contributions:** Conceptualization, A.R.; methodology, A.R.; software, A.R.; validation, A.R.; formal analysis, A.R.; investigation, A.R., M.F.; resources, A.R., M.F.; data curation, A.R., M.F.; writing—original draft preparation, A.R.; writing—review and editing, A.R.; visualization, A.R., M.F.; supervision, M.F.; project administration, M.F.; funding acquisition, M.F. All authors have read and agreed to the published version of the manuscript.

**Funding:** This research was funded by NATURAL SCIENCES AND ENGINEERING RESEARCH COUNCIL OF CANADA (NSERC), grant number RGPIN-2020-06979.

**Institutional Review Board Statement:** Not applicable.

**Informed Consent Statement:** Not applicable.

**Data Availability Statement:** Not applicable.

**Conflicts of Interest:** The authors declare no conflict of interest.

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
