# Peer review of "Investigation of Air Pocket Behavior in Pipelines Using Rigid Column Model and Contributions of Time Integration Schemes"

_water, doi:10.3390/w13060785_

Round 1

Reviewer 1 Report

The paper is very interesting. The structure of the paper is correct. First, introduction and bibliographic review. Then, mathematical model and presentation of results. Finally, analysis of results and conclusions. Nevertheless, some aspects of the paper should be revised and improved:

  • The authors say in the abstract: “The paper studies the air pressurization problem caused by a partially pressurized transient flow in a reservoir-pipe system. The purpose of this study is to analyse the performance of the rigid column model in predicting the damping of the air pressure distribution”. In my opinion, the title of the paper should be more specific. The title should include some reference on entrapped air pocket or something similar.
  • The bibliographic review carried out in the introduction should be more exhaustive. There are several research groups in Portugal, Spain, Italy, Holland, USA, China, etc., which have worked on topics related to the problem of entrapped air in pipes. Many of these works appear in the paper: “Hydraulic modeling during filling and emptying processes in pressurized pipelines: a literature review”. Urban Water Journal. 2019, 16(4), 299-311.
  • The authors say in section 2.1: “In this paper, the air entrapment caused by transient flow in a frictionless, horizontal reservoir-pipe system is studied”. “Note that for simplicity of the discussion the friction and local loss terms are neglected”. This simplification is very strong. If you consider a frictionless system, is this hypothesis realistic? The authors should adequately justify this question.
  • Line 125: “However, commonly in the literature, the effect of heat transfer is neglected”. This depends on the characteristics of the system and the characteristics of the transient.
  • Line 185: “From the literature, it can be realized that classical 4th order Runge-Kutta scheme, hereafter called the 4th-RK scheme, has been used to integrate the governing equations of the RC model”. The authors should add some reference.
  • Line 226: “The physical dissipation of the heat transfer can be compensated by the numerical dissipation of the BE scheme”. Is this verified?
  • Many simplifications have been made to obtain the mathematical equations. Before presenting the results in section 3 of the paper, all the simplifications made should be perfectly clear. It should be clear in which cases this methodology can be applied.
  • All graphics should be in color. In this way, the differences between the drawn curves would be much better appreciated.
  • In my opinion, section 4 "Discussion" could be integrated into section 5 "Conclusions".
  • The authors should clearly state in the conclusions the limitations of the proposed methodology due to the simplifications made.
  • The bibliographic review is very brief. There are only three references from the last 5 years.

To conclude, in my opinion, the paper can be accepted with modifications, mainly related to complete and clarify some issues. The authors should also complete the literature review.

Reviewer 2 Report

The reviewer has found the paper to be well written and the motivation of the work was clearly defined. The authors have developed an improvement to the lumped model for predicting air/water interaction under unsteady conditions. In particular, the authors propose that the simplicity of the lumped inertia model can be partially overcome by using a numerical dispersive scheme and using the numerical dispersion to take into account actual physical attenuation effects. The validation of the work was carried out numerically as well as experimentally using deviously published data. From a scientific approach standpoint, the reviewer cannot fault the paper as the author has taken care to validate their methodology and the arguments were well presented.

The reviewer’s main concern is with the fundamental premise of the paper. The lumped inertia mode is a very simple model that can only be applied under the simplest of physical cases. Apart from the lack of frictional dispersive effects and heat transfer, the core model assumes that the fluid column act as one single unit which is only a realistic assumption when the pipe is short. The authors have demonstrated their method on very short (<10m) pipeline experiments  but it is clear that when the pipeline contains changes in elevation, junctions, large diameter or when the wave travel time within the water column approaches the compressive time scale of the air pocket that the model will fail to capture the dynamics of the entire pipeline system.

In the reviewer’s opinion, it is critical for the authors to explain under what type of physical situations would their model be of benefit. It seems to the reviewer that the model is effectively trying to offer a slight improvement to arguably the simplest  model when more comprehensive and accurate approaches are available to the readers. To make matters worse, the model uses a non-physical approach to introduce damping to the results which does not sit well with the reviewer. To intentionally solve a set of equations poorly by using a coarse discretisation in an effort to simulate physical attenuation effect will certainly fail once the complexity of the system increase. The reviewer was left with the question of why the paper is necessary and the scientific novelty of the paper is unclear. For this reason, the reviewer recommend that the authors make efforts to answer the issues of scientific novelty before it can be accepted for publication.

Round 2

Reviewer 1 Report

After this last review, the authors have made changes and the paper has been improved significantly. The paper can be published.

Reviewer 2 Report

The authors have made effort to clarify the queries identified in the last review. The paper can be accepted in its current form.